# Constructing Taxonomies: Identifying Distinctive Class of HIV Support and Risk Networks among People Who Use Drugs (PWID) and Their Network Members in the HPTN 037 Randomized Controlled Trial

**DOI:** 10.3390/ijerph19127205

**Published:** 2022-06-12

**Authors:** Laurens G. Van Sluytman, Carl Latkin, Laramie R. Smith

**Affiliations:** 1School of Social Work, Morgan State University, Baltimore, MD 21251, USA; 2Department of Health, Behavior and Society, Bloomberg School of Public Health, Johns Hopkins University, Baltimore, MD 21218, USA; carl.latkin@jhu.edu; 3Division of Global Public Health, UCSD School of Medicine, University of California, San Diego, CA 92121, USA; lrs003@ucsd.edu

**Keywords:** people who use injection drugs (PWUID), taxonomy, paraphernalia, latent class, analysis, HIV

## Abstract

Injection drug use is a significant mode of HIV transmission. Social networks are potential avenues for behavior change among high-risk populations. Increasing knowledge should include a classification or taxonomy system of networks’ attributes, risks, and needs. The current study employed 232 networks comprising 232 indexes, with 464 network members enrolled in Philadelphia. LCA revealed a three-class solution, Low-Risk, Paraphernalia Risk, and High Sex/Moderate Paraphernalia Risk class, among participants. The analysis found receiving money or drugs for sex and employment status increased the odds of belonging to PR and PSR classes. Homelessness and incarceration increased the odds of belonging to the PR class when compared to the LR class. Our findings suggest that classes of risk among PWID comprise clusters of information concerning their members. These findings add depth to our understanding while extending our knowledge of the contextual environment that nurtures or exacerbates the problem.

## 1. Introduction

An estimated 1.1 million people are currently living with HIV infection in the United States. Of these, 166,000 (15%) have not yet been diagnosed [1]. Injection drug use continues to be a significant mode of transmission [2]. The CDC [3] also reports that in 2015, approximately 6% (2392) of the 39,513 diagnoses of HIV and 10% (1804) of the 18,303 AIDS diagnoses were attributable to injection drug use. Additionally, a significant amount of HIV transmission among people who inject drugs (PWID) occurs due to an increased likelihood of engaging in unprotected sexual intercourse [4,5]. Behaviors associated with reducing the risk of transmission include using clean syringes and other paraphernalia (i.e., cookers, cottons, and water) [6]. However, Lea and colleagues and others [7,8,9] reported that sexual risk among PWID remains a challenge. Targeted intervention, treatment, and care with people who inject drugs (PWID) require substantial attention [10] and evidence-based methods of engaging and training PWID.

Social networks, crucial building blocks of communities, are a potentially useful avenue for promoting behavior change among high-risk populations [11,12,13,14,15,16]. Individuals in small groups who share a common interest, friendship, knowledge, and other forms of interdependency may work to resolve their network members’ problems and influence each other’s behaviors [17,18,19]. 

More specifically, researchers [20,21,22,23,24] have examined the impact of social networks on HIV risk behavior among various populations: adolescents [25], African American women at risk of heterosexually acquired HIV [26], immigrants [27,28], men who have sex with men (MSM) [29], sex industry workers [30], and transgender women [31]. These researchers found that social networks have the potential to change social norms and behaviors among at-risk groups. However, networks may vary in critical domains [32]. For example, gender differences [20] and size [33,34] in social network influence risk behaviors among injection drug users. These authors suggest that these findings offer support for gender-specific prevention strategies and highlight the need to identify different types of networks to target naturally occurring risk groups more effectively. Further network analyses are usually conceptualized as the network attributes associated with an individual, such as the density or attributes of a whole network. Additionally, they frequently provide information on how individuals are linked together but often do not examine underlying network attributes that may distinguish networks. A novel approach to increasing the depth of research and providers’ knowledge base should include a system of classification or taxonomy that offers insight into attributes to identify networks, their various risks, and associated needs.

### 1.1. Background

#### Taxonomies

As early as 1978, Fischhoff, Slovic, Lichtenstein, Read, and Combs [35] used psychometric procedures to quantitatively assess the perceived risk, acceptable risk, and perceived benefit of 30 activities and technologies. They found that reducing risk to acceptable levels was correlated with perceived benefit. Building upon this work, Slovic, Fischhoff, and Lichtenstein [36] developed a taxonomy for risk based on the perception of risk associated with activities, disease, substances, technologies, and desire for risk reduction and regulation. More recently, Marsch, Bickel, Badger, and Quesnel [37] employed Slovic and colleagues’ taxonomy of risk scale to examine perceptions of risk deaths among Intravenous Drug Users (IDUs). The authors identified differential response clustering patterns among IDUs and their control counterparts. For example, IDUs rated items such as weapons and alcoholic beverages as riskier than control participants. Though the research employed cross-sectional data, the authors endorsed the necessity of examining differences among IDU populations, employing longitudinal methods to enable an increased understanding of the high-risk behavior among this group. To date, there is no agreed-upon method for classifying networks’ characteristics of PWID. Classifying the interaction of co-occurring individual network factors and norms among at-risk populations would advance our knowledge of communities and increase opportunities for developing targeted interventions [38]. Public health acknowledges the importance of homophily. For example, among Black MSM, racial homophily, while reducing network size, may be a critical driver of disparities in HIV [39]. However, Matthews and colleagues [40] caution that the ties that bind these networks are predicated on factors that go beyond the disease to shared experiences, social adversities, and necessity. Thus, interventions targeting health disparities should not seek to disperse the network but rather identify its strength. For example, Christakis and Fowler [41] found homophily to be important in information sharing among cigarette smokers. Similarly, Centola [42] identified that homophily influences the early adoption of new health behaviors among those most in need. However, factors comprising information needs or variations among those population most in need at the network level remain elusive.

Identifying taxonomies among networks would also significantly increase the capacity to identify and predict variations among groups and target specific groups for interventions, and model how behaviors and infectious diseases may spread in different types of groups. Such a taxonomy would provide an integrative framework to identify the correlations of individual, community, and structural risk factors. Targeted interventions may focus on the individual, community, and broader structural factors that shape the network characteristics, such as disparities in housing, under- or unemployment, and exposure to discrimination, which contribute to risk.

This study aimed to improve scientific knowledge, technical capability, and clinical practice by identifying taxonomies. The study conceptualized taxonomy as the interaction of individual and social membership and norms to classify members within heterogeneous HIV support and risk networks [43] into distinct subgroups. It intended to: (1) identify taxonomies of networks among HIV-negative people who use drugs (PWID) and members of their HIV support and risk networks, (2) determine the characteristics of the network, and (3) determine the impact of the individual and network-level factors on class membership.

## 2. Data and Methods

This study was a secondary data analysis of HPTN 037: A Phase III randomized study to evaluate the efficacy of a network-oriented peer education intervention to prevent HIV transmission among PWID and their network members at two sites (Philadelphia, PA, USA, and Chiang Mai, Thailand). The study employed a diverse theoretical frame (e.g., diffusion of information, social learning, role theory, cognitive dissonance, and social norm) to present risk reduction strategies to network peer educators. The study was approved by the Institutional Review Board (IRB) at Johns Hopkins University, University of Pennsylvania, Chiang Mai University, and the Thailand Ministry of Public Health [44]. Latkin and colleagues [44] reported sharp reductions in injection risk behaviors between baseline and follow-up and found no change in sexual risk due to the intervention. The current study drew its sample from the 232 networks enrolled in Philadelphia, comprising 232 indexes with 464 network members. Outreach workers recruited index participants. Index” participants met the following eligibility criteria: aged 18 or older, injected drugs at least 12 times in the prior three (3) months, not enrolled in methadone maintenance treatment in the past three (3) months, HIV-negative antibody test results within 60 days before randomization, willingness to identify and attempt to recruit at least two eligible HIV risk network members and recruit at least one eligible risk network member. Network member eligibility criteria include: aged 18 or older, recruited for the study by an eligible index participant, and injected drugs with or had sex with the index participant who recruited them within the previous three months. It employed data collected at baseline and a 6-month follow-up from both index and network members at baseline and follow-up interviews.

### 2.1. Measures

This project conceptualized risk as engaging in behavior associated with HIV transmission, e.g., sharing rinse water or other drug use paraphernalia. Injection risk was assessed using the following questions: “In the last month, how many times did you use rinse water that others had used?”, “use a cooker that others had used?”, “use cotton that others had used?”, “inject drugs that were frontloaded or backloaded into the syringe or needle that you used?”, “use a needle that others had discarded?”, “use a needle or syringe after someone that you know is HIV-positive used it?”, and “did you ever inject drugs with others in a shooting gallery, in an abandoned building, in a car, or in a public park or public restroom?”. These continuous variables were collapsed to produce a dichotomous variable where 0 indicated “Never/No” and 1 indicated “One or more time/Yes”. Participants were asked, “In the last month, did you ever clean your needle either before or after injecting?” to assess needle cleaning practice. Sexual risk comprised two categories of partners, primary and other partners. Data collection included asking participants, “how many times in did you have vaginal or anal sex with your partner, and how many of these times did you (or your partner) use a condom?” All responses were coded as 0 for “Yes”, or 1 for “No”.

### 2.2. Analysis

The analysis used data from the baseline visit and six-month follow-up. Preliminary analyses used the statistical software Mplus 7.0 and IBM SPSS Statistics 20 (IBM, Armonk, NY, USA). The statistical analyses of data included univariate analysis to measure the distribution of all the variables used in this study within the sample population. Next, latent class analysis (LCA) was conducted using Mplus, Version 7.0 [45], to explore subgroup heterogeneity among PWID. Latent class indicators included injection drug use and sexual risk behavior variables (e.g., sharing cookers and unprotected sexual intercourse with a primary partner). To determine the number of groups that were best represented by the data, three criteria were considered [46,47]: (1) Bayesian information criterion (BIC), in which smaller values of BIC indicate a better fit model, (2) Akaike’s information criterion (AIC), such that the model with the fewest parameters is better, and (3) that entropy, an index for assessing the precision of assigning latent class membership, is greater than 0.80. The analysis then used binary and multinomial logistic regression to examine the association between individual and network latent class membership and demographics and compare the likelihood of class membership within and across classes. In both cases, *p*-values at or below 0.05 were considered to indicate statistical significance.

## 3. Results

This study’s analyses included 232 index HIV-negative PWID and members of their HIV support and risk networks. The 232 index participants reported a total of 464 social network members comprising 451 network members. Each social network group comprised an average of two social network members. One network included only the index. This network was removed from the analysis. Table 1 and Table 2 present the demographic characteristics of index and network members and networks, respectively.

Among the 232 index participants, those between the ages of 30 and 39 and 40 and 49 represented the largest groups, at 26 and almost 37 percent, respectively. This was also true of network members who were almost 30 and 34. Male participants represented almost 80 percent of index participants and 63 percent of network members. Forty-five percent of index members were Non-Latino Black, followed closely by Non-Latino White at 40 percent. For network members, each group represented 45.5 percent of the sample. Forty-seven percent of index and network member participants reported having completed high school or GED programs. Though index participants reported high rates of unemployment, 77 percent, only 21 percent (each) reported being homeless or incarcerated during the previous six months. Similarly, 38 percent of network members reported being unemployed, 26 percent being homeless or incarcerated, and 15 percent during the last six months. 

Table 2 describes the demographic characteristics of the 231 networks—approximately 73 percent of networks comprised at least three members. Exclusively or predominantly Black networks represented almost 40 percent of the sample, followed by exclusively or predominantly White networks at 36 percent. Few networks comprised exclusively or predominantly female participants, at almost 17 percent. 

### 3.1. Taxonomies

LCA revealed a three-typology solution among participants. Model fit was determined by an Akaike information criterion of 6130.05, Bayes information criterion of 6288.42, and entropy of 0.804. Presented below are the model fit criteria outcomes for three, four, and five classes.

Table 3 presents membership and item response probability are depicted in Figure 1. The three derived risk latent classes were Low-Risk, Paraphernalia Risk, and Paraphernalia/Sex Risk. The Low-Risk class comprised the greatest number of networks, 382 (41%). Low levels of IDU risk, e.g., shared water, cookers, cottons, and use of shooting galleries, characterized this class. They responded affirmatively to lower rates of using discarded needles and lower rates of engaging in high-risk sexual activity with primary and other partners than other groups.

Paraphernalia Risk was the second most prevalent class, at 237 (35%). High levels of IDU risk characterize this type, e.g., shared cookers. They tended to respond affirmatively to using discarded needles; these rates remained higher than those in Low-Risk networks. Additionally, their rates of engaging in high-risk sexual activity with primary were higher than member of the Low-Risk class.

Participants in the Paraphernalia/Sex Risk class comprised the smallest number of respondents, 162 (24%). Moderate levels of IDU risk in all risk categories, i.e., shared cookers, characterized this class. As with the Paraphernalia Risk class, they tended to respond affirmatively to using discarded needles and using needles after another. However, these rates remained higher than those in the Low-Risk networks. Their rates of engaging in high-risk sexual activity with primary sex partners were higher than both Low-Risk and Paraphernalia Risk classes.

### 3.2. Class Membership

Table 4 presents the outcomes of logistic regression analysis of selected individual and network characteristics for the odds of belonging to the three risk classes—Low-Risk, Paraphernalia Risk and Paraphernalia, and Sex Risk classes. Several individual and demographic factors significantly impacted the odds of membership in the Low-Risk class. They include age, engaging in exchange sex, network size, and racial composition. Accordingly, the odds of membership in this class increased among individuals aged 40 to 49 years old, and 50 years and older (OR = 1.39; 95% CI 0.86, 2.23) and (OR = 1.79; 95% CI 1.05, 3.04), respectively. However, findings were statistically significant only for those 50 years and older. Odds of membership in this class reduced significantly if one engaged in exchange sex, either giving (OR = 0.45; 95% CI 0.20, 1.00) or receiving (OR = 0.15; 95% CI 0.02, 1.25) money or drugs for sex. Network demographic characteristics, i.e., belonging to a network comprising exclusively or predominantly Black members (OR = 1.35; 95% CI 0.72, 2.52) increased the odds of membership in the High Sex/Moderate Paraphernalia-Risk class.

### 3.3. Class Comparison

Multinomial logistic regression, presented in Table 5, compared the odds of belonging to the Paraphernalia Risk and Paraphernalia and Sex Risk classes to the Low-Risk class based on individual and network characteristics. The analysis found receiving money or drugs for sex to be a significant factor increasing the odds of belonging to the two risk classes, Paraphernalia Risk class (OR = 7.75; 95% CI 2.34, 25.66) and Paraphernalia and Sex Risk class (OR = 4.34; 95% CI 1.16, 16.25). Further, giving money or drugs for sex within the last month increased the odds of belonging to the High Sex and Moderate Paraphernalia class (OR = 2.82; 95% CI 1.1, 7.21). Among network composition factors, belonging to a predominantly Black network increased the odds of belonging to the High Sex and Moderate Paraphernalia class (OR = 1.08; 95% CI 1.08, 9.73).

## 4. Discussion

Green and McDermott [48] argued that making use of findings of traditional scientific inquiry to intervene in social problems requires incorporating the “knowledge of the structure and function of our world” (p. 2414). Further, the importance of the social environment to human well-being is well-established. The social environment comprises the individual’s relationship with others in the immediate environment and communities they belong to. The social environment also comprises organizational and structural factors that determine human behavior [49]. It is also well-documented that the individual’s access to resources mediates the social environment. The acquisition of resources may be hampered by multiple forms of discrimination based on age, gender, or race or facilitated by membership in various social networks [50].

Additionally, members of a high-functioning network may interact exclusively with other high-functioning individuals, leaving less well-functioning individuals to interact predominantly with each other [42,51,52]. For these reasons, building and testing effective interventions must incorporate knowledge of variations within target populations. Furthermore, while avoiding notions of monolithic populations, empirical research must strive to identify patterns among populations that may be informed by numerous factors integral to the individual, the networks to which they belong, and macro-level forces informing access to resources, thereby increasing our knowledge base and scope of practice behaviors.

Though the data were collected to assess the efficacy of a peer education intervention, our findings advance our knowledge in developing an IDU taxonomy in paraphernalia and sexual risk behavior and the impact of individual and network-level factors. Given the sample size and recruitment methods, these findings may be generalizable to larger populations of similarly affected individuals and their networks. Although we identified three distinct classes within this sample and factors associated with belonging to these classes, the implications of our findings include the need for effective assessment and targeted intervention, including behavioral and policy-driven interventions.

First, our findings demonstrate that there were several distinct classes within the sample. In contrast to the class of participants who reported neither paraphernalia nor sex risk, the two remaining classes demonstrated that PWID may have reduced sex risk while struggling with the risk associated with paraphernalia in the Paraphernalia Risk class or continuing to struggle with both, as those in the Paraphernalia and Sex Risk class do. Equally important to the classification is the identification of paraphernalia risk in both classes involving cookers.

We further identified that multiple individual-level factors, such as exchanging sex, age, gender, race, and network-level factors such as the network’s size, gender, and racial composition, play important roles in informing class membership. For example, participants 50 years old and above, or Latino/Hispanic individuals, were more likely than their younger counterparts, aged 18 to 29 years old, to belong to the highest risk class, i.e., those who engage in both high-risk IDU behavior and unprotect intercourse. In fact, Latino participants were more than two times more likely to belong to this class than their White counterparts. Though beyond this paper’s scope, these differences may signal the need to produce intervention material that reflects various needs across populations. In such interventions, culture comprises not only race and ethnicity but age. In addition to these individual-level factors, we found network composition, such as networks comprising five or more members, are less likely to belong to either the Paraphernalia Risk or Paraphernalia and Sex Risk class. However, networks comprising exclusively or predominantly Black members faced significantly higher odds of belonging to the Paraphernalia and Sex Risk class. We also found that the impact of the networks’ average age contributed to their likelihood of belonging in these two risk classes. For example, those 40 years and above were less likely to belong to the Paraphernalia-Risk-only class than their younger counterparts. However, in the Paraphernalia and Sex Risk class, only those 50 years old and above maintained their reduced odds.

These variations and the finding that various risks are not independent of each other point to the need for assessments beyond the simple accrual of psychosocial demographic and risk behavior material. Graybeal [53] argued that providers often rely on a set of “problem- and pathology-based assessment forms” (p. 234). He argues in favor of models that affirm the perspective that clients “holds the clues and creativity that will lead to solutions” (p. 241).

As such, our findings suggest that classes of risk among PWID comprise clusters of information concerning their members. For example, Black participants were more likely to belong to higher risk IDU and sexual risk classes, though not exclusively. The literature has demonstrated the relationship between exposure to racism and psychological distress and the association between psychological distress and high-risk behaviors. Given the current political climate and increased rates and visibility of instances of blatant violence and discrimination against minority and immigrant populations [54,55,56], assessments must make inquiries into the broader context of social problems and challenges. Similarly, while women and older individuals are more likely to belong to the Low-Risk class, both groups faced high unemployment rates in our sample. Our findings suggest that unemployment increases the odds of belonging to high-risk classes. In fact, in this sample, exclusively or predominantly female networks were more likely to belong to the Paraphernalia Risk class. This class also favored significant odds for those engaged in exchange sex—specifically those who received money or drugs, in contrast to the class engaged in both paraphernalia and sex risk. Almost 60 percent of the networks in this class were predominantly or exclusively male. For this class, the risk associated with receiving money or drugs reduced the likelihood of belonging. Instead, the likelihood of belonging to this class was two times higher for those who gave money or drugs for sex. Given the predominance of men in this network, it may be posited that the purchaser may have considerable control over the level of risk involved in the sex exchange. These findings add depth to our understanding while extending our knowledge of the social problem. The effectiveness of interventions would increase as they become more responsive to the contextual environment that nurtures or exacerbates the problem or pathology.

Another implication of these findings concerns policy as intervention and evaluation of present policy. In 2015, the Consolidated Appropriations Act, 2016 (Pub. L. 114-113) [57] included the provision of sterile needles, syringes, and other drug preparation equipment (purchased with non-federal funds) among the “appropriate” services to be offered by syringe exchange programs (SEPs) or needle exchange programs (NEPs). The Centers for Disease Control (CDC) [4] described the span of services provided by SEPs or NEPs to reduce new HIV infections among PWID. Furthermore, though needle exchange has demonstrated a substantial impact on reducing HIV transmission among PWID, halting transmission remains a challenge [58]. Our findings suggest that though participants reported low levels of risk associated with sharing needles, one class struggled with or chose not to implement this intervention.

Further, in the two risk classes, both reported sharing cookers. Distributing cookers is not included among the services described by the CDC. Thus, the need for a policy that transparently articulates the needs of at-risk communities remains elusive. In addition, the individual (e.g., age and gender) and network (size and racial composition) reveal both challenges and opportunities. For example, addressing the structural and physical barriers that may limit access to services for female PWID who face particular risk concerning exchange sex, Latinos, or networks that comprise exclusively or predominantly Black PWID who have faced exposure to discrimination. At the same time, our awareness that larger network sizes may reduce risk offers us an opportunity to design interventions for small groups and dyads.

Our data face several limitations. As a secondary data analysis, the key measures were not specifically designed to examine the research questions proposed by this study. Accordingly, there were gaps in information that could add substantially to the depth of the analysis. For example, delving into the nature of the relationship among network members, beyond whether they were sexual partners or not, informed by social network exchange theory, would add greater depth to the analysis. For instance, does having one’s spouse in one’s network increase the likelihood of condom use with other partners while increasing the risk of other high-risk activities such as backloading or using after their partner? El-Bassel and colleagues [59] reported that many rely on their partner to secure the drug and inject after their partner for complicated reasons, including fear of physical or sexual violence. Another limitation recognizes that individuals may belong to multiple networks over time. As needs change, the individual may seek resources from other networks.

## 5. Conclusions

Social networks are living organizations. Therefore, they are responsive to numerous transformations in the social environment. As such, questions concerning network affiliation over time would have added substantial depth to our analysis. Despite these limitations, networks are relationships established to facilitate exchanges of resources. Cook, Cheshire, Rice, and Nakagawa, [60] and Lawler, Yoon, and Thye [61], stated that network members develop relational cohesion to maintain the relationship. Through maintenance, the relationship may encompass meeting more than the initial goals of the original engagement. Instead, they may extend to new ventures. Network members whose class matches that of their index members have greater odds of belonging to the Low-Risk class than the Paraphernalia and Sex Risk class. Thus, the relationship of matching classes between the index and network members may offer greater information exchanges. These similarities in class may extend beyond injection drug use and sexual behavior to encompass accessing other needed resources. This being the case, understanding networks may offer the opportunity to engage members in creating genuinely participatory and emancipatory interventions in very austere times.

## Figures and Tables

**Figure 1 ijerph-19-07205-f001:**
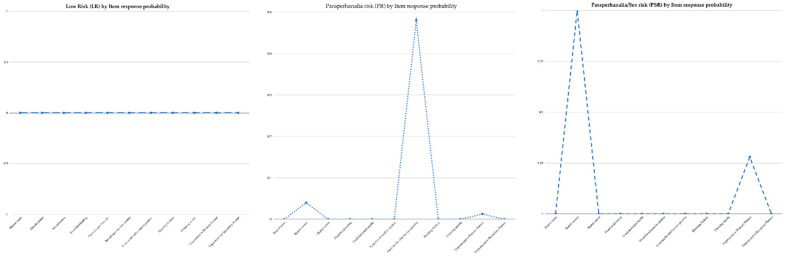
Latent class categories.

**Table 1 ijerph-19-07205-t001:** Demographic characteristics of index participants (*n* = 232) by latent class.

			Total		Low- Risk Class		Paraphernalia Risk Class		High Sex/Moderate Paraphernalia Risk Class	
			*n*	%	*n*	%	*n*	%	*n*	%
Age										
	<29 years old		43	18.6	13	14.8	20	24.4	10	16.4
	30–39 years old		61	26.4	21	23.9	23	28.0	17	27.9
	40–49 years old		85	36.8	34	38.6	29	35.4	22	36.1
	50 years old+		42	18.2	20	22.7	10	12.2	12	19.7
Gender										
	Male		184	79.7	74	84.1	60	73.2	50	82.0
	Female		47	20.3	14	15.9	22	26.8	11	18.0
Race										
	Non-Latino White		93	40.3	35	39.8	33	40.2	25	41.0
	Non-Latino Black		104	45.0	41	46.6	38	46.3	25	41.0
	Latino		22	9.5	8	9.1	6	7.3	8	13.1
	Other		12	5.2	4	4.5	5	6.1	3	4.9
Education										
	<High School		75	32.5	29	33.0	27	32.9	19	31.1
	High School/GED		109	47.2	43	48.9	36	43.9	30	49.2
	Some College+		47	20.3	16	18.2	19	23.2	12	19.7
Employment	Employment									
	Unemployed		178	77.1	65	73.9	68	82.9	45	73.8
	Part-time, Occasional or time to time		31	13.4	12	13.6	9	11.0	10	16.4
	Full time (≥30 h per week)		22	9.5	11	12.5	5	6.1	6	9.8
Homeless (last 6 months)										
	No		181	78.4	70	79.5	64	78.0	47	77.0
	Yes		50	21.6	18	20.5	18	22.0	14	23.0
Incarceration (last 6 months)										
	No		182	78.8	72	81.8	60	73.2	50	82.0
	Yes		49	21.2	16	18.2	22	26.8	11	18.0
Network Member										
Age										
	<29 years old	70	15.5	25	13.7	35	19.3	10	11.5	70
	30–39 years old	135	29.9	46	25.1	63	34.8	26	29.9	135
	40–49 years old	154	34.1	67	36.6	57	31.5	30	34.5	154
	50 years old+	92	20.4	45	24.6	26	14.4	21	24.1	92
Gender										
	Male	287	63.6	123	67.2	110	60.8	54	62.1	287
	Female	164	36.4	60	32.8	71	39.2	33	37.9	164
Race										
	Non-Latino White	205	45.5	82	44.8	79	43.6	44	50.6	205
	Non-Latino Black	205	45.5	88	48.1	81	44.8	36	41.4	205
	Latino	20	4.4	5	2.7	11	6.1	4	4.6	20
	Other	21	4.7	8	4.4	10	5.5	3	3.4	21
Education										
	<High School	153	33.9	54	29.5	69	38.1	30	34.5	153
	High School/GED	215	47.7	91	49.7	81	44.8	43	49.4	215
	Some College+	83	18.4	38	20.8	31	17.1	14	16.1	83
Employment	Employment									
	Unemployed	375	83.1	152	83.1	152	84.0	71	81.6	375
	Part-time, Occasional or time to time	39	8.6	15	8.2	15	8.3	9	10.3	39
	Full time ≥ 30 h per week)	37	8.2	16	8.7	14	7.7	7	8.0	37
Homeless (last 6 months)										
	No	332	73.6	124	68.1	141	77.9	67	77.0	332
	Yes	118	26.2	58	31.9	40	22.1	20	23.0	118
Incarceration (last 6 months)										
	No	381	84.5	155	85.2	154	85.1	72	82.8	381
	Yes	69	15.3	27	14.8	27	14.9	15	17.2	69

**Table 2 ijerph-19-07205-t002:** Demographic characteristics of networks (*n* = 231 *) by latent class.

		Total		Low- Risk Class		Paraphernalia Risk Class		Paraphernalia & Sex Risk Class	
		*n*	%	*n*	%	*n*	%	*n*	%
Size									
	2 members	60	26.0	24	27.3	22	26.8	14	23.0
	3 members	84	36.4	34	38.6	25	30.5	25	41.0
	4 members	33	14.3	12	13.6	12	14.6	9	14.8
	5+ members	54	23.4	18	20.5	23	28.0	13	21.3
Age									
	18–29 years old	31	13.5	11	12.6	12	14.6	8	13.1
	30–39 years old	75	32.6	25	28.7	31	37.8	19	31.1
	40–49 years old	88	38.3	31	35.6	31	37.8	26	42.6
	50 years old+	36	15.7	20	23.0	8	9.8	8	13.1
Race									
	Exclusive or Predominant White	84	36.4	32	36.4	30	36.6	22	36.1
	Racially Equal	54	23.4	15	17.0	21	25.6	18	29.5
	Exclusive or Predominant Black	92	39.8	40	45.5	31	37.8	21	34.4
Gender									
	Predom. or Exclusively Female	38	16.5	17	19.3	14	17.1	7	11.5
	Equal Male and Female	51	22.1	19	21.6	13	15.9	19	31.1
	Predom. or Exclusively Male	142	61.5	52	59.1	55	67.1	35	57.4

* One network comprising one member was removed from the analysis.

**Table 3 ijerph-19-07205-t003:** Model fit indices by latent class.

	3 Classes	4 Classes	5 Classes
Akaike (AIC)	6130.05	6109.029	6099.145
Bayesian (BIC)	6288.42	6321.706	6366.121
Entropy	0.804	0.725	0.705

**Table 4 ijerph-19-07205-t004:** Logistic regression analysis of selected demographic variables on latent class membership. (*n* = 696).

				Low- Risk Class		Paraphernalia Risk Class			Paraphernalia and Sex Risk Class		
		OR		95% C.I.	*p*	OR	95% C.I.	*p*	OR	95% C.I.	*p*
Age	<29 years old	1.00	Ref			1.00			1.00		
	30–39 years old	1.00		0.61, 1.64	1.00	0.83	0.52, 1.34	0.45	1.33	0.73, 2.41	0.35
	40–49 years old	1.39		0.86, 2.23	0.18	0.60	0.38, 0.95	*	1.34	0.75, 2.39	0.33
	50 years old+	1.79		1.05, 3.04	*	0.40	0.23, 0.68	*	1.54	0.82, 2.92	0.18
	female	1.25		0.88, 1.78	0.21	0.80	0.57, 1.13	0.21	1.00	0.66, 1.51	0.99
Race	Non-Latino White	1.00	Ref		0.54	1.00		0.73	1.00		0.51
	Non-Latino Black	1.09		0.78, 1.51	0.63	1.06	0.76, 1.49	0.73	0.82	0.56, 1.22	0.33
	Latino	0.66		0.33, 1.34	0.25	1.19	0.6, 2.34	0.62	1.32	0.64, 2.74	0.46
	Other	0.84		0.39, 1.79	0.65	1.51	0.72, 3.17	0.28	0.72	0.28, 1.84	0.50
Education	<High School	1.00	Ref			1.00			1.00		
	High School/GED	1.21		0.85, 1.72	0.28	0.78	0.55, 1.1	0.15	1.06	0.71, 1.6	0.77
	Some College+	1.23		0.79, 1.92	0.35	0.86	0.55, 1.33	0.50	0.91	0.54, 1.56	0.74
Employment	Full-Time Employment	1.00				1.00			1.00		0.55
	Unemployed	0.77		0.45, 1.33	0.36	1.38	0.77, 2.45	0.28	0.94	0.49, 1.81	0.85
	Part-time, Occasional or time to time	0.78		0.38, 1.58	0.49	1.07	0.51, 2.25	0.86	1.29	0.57, 2.92	0.55
Homelessness	No	1.00	Ref			1.00			1.00		
	Yes	1.34		0.94, 1.91	0.11	0.79	0.55, 1.14	0.20	0.91	0.59, 1.41	0.68
Incarceration	No	1.00				1.00			1.00		
	Yes	0.89		0.59, 1.34	0.57	1.11	0.74, 1.67	0.61	1.01	0.62, 1.64	0.96
Exchange Sex/Interaction											
Gave Money/Drug for Sex (last month) (1)	No	1.00	Ref			1.00			1.00		
	Yes	0.45		0.2, 1	*	1.09	0.5, 2.35	0.83	2.16	0.98, 4.76	0.06

* *p* < 0.05.

**Table 5 ijerph-19-07205-t005:** Multinomial logistic regression analysis of selected individual and network demographic characteristics variables for latent class membership. (*n* = 696).

		Low- Risk Class ^a^		Paraphernalia Risk Class			High Sex/Moderate Paraphernalia Risk Class		
				OR	CI	*p*	OR	CI	*p*
Age	<29 years old	1	Ref						
	30–39 years old			0.52	0.15, 1.82	0.31	1.21	0.27, 5.39	0.8
	40–49 years old			0.52	0.14, 1.96	0.34	0.85	0.18, 4.05	0.84
	50 years old+			0.24	0.05, 1.31	0.10	0.6	0.1, 3.68	0.58
Gender	Female	1	Ref						
	Male			1.09	0.34, 3.52	0.89	0.72	0.19, 2.69	0.63
Race	Non-Latino White								
	Non-Latino Black			0.9	0.39, 2.08	0.81	0.61	0.24, 1.58	0.31
	Latino			0.97	0.23, 4.12	0.97	2.57	0.63, 10.51	0.19
	Other			0.96	0.13, 7.17	0.97	0.45	0.05, 4.37	0.49
Education	<High School	1	Ref						
	High School/GED			0.73	0.28, 1.88	0.52	0.80	0.28, 2.33	0.68
	Some College +			1.05	0.36, 3.08	0.93	0.77	0.23, 2.59	0.68
Employment	Full time (≥30 h per week)	1	Ref						
	Part-time, Occasional or time to time			1.14	0.22, 5.81	0.87	1.6	0.33, 7.64	0.56
	Unemployed			2.66	0.71, 9.94	0.14	1.54	0.41, 5.75	0.52
Homelessness	No	1	Ref						
	Yes			1.24	0.48, 3.2	0.66	1.16	0.39, 3.47	0.80
Incarceration	No	1	Ref						
	Yes			1.01	0.37, 2.72	0.99	0.71	0.22, 2.25	0.56
Gave Money/Drug for Sex (last month)	No	1	Ref						
	Yes			2.17	0.87, 5.4	0.10	2.82	1.1, 7.21	*
Received Money/Drug for Sex (last month)	No	1	Ref						
	Yes			7.75	2.34, 25.66	**	4.34	1.16, 16.25	*
**Network Composition**									
Index—Network Member Latent Class Match									
	No	1	Ref						
	Yes			1.35	0.89, 2.04	0.16	0.43	0.48, 1.37	0.43
Index—Network Member Latent Class Match									
Network Size	≤2 member	1	Ref						
	3 members			1.11	0.72, 1.71	0.63	1.03	0.62, 1.72	0.89
	4 members			1.12	0.65, 1.93	0.69	1.19	0.63, 2.23	0.60
	5+ members			0.99	0.6, 1.63	0.98	0.98	0.55, 1.77	0.95
Network Age Composition	≤29 years old	1	Ref						
	30–39 years old			1.01	0.32, 3.18	0.99	1.42	0.36, 5.71	0.62
	40–49 years old			0.85	0.26, 2.75	0.79	1.86	0.47, 7.35	0.37
	50+ years old			0.28	0.07, 1.11	0.07	0.27	0.05, 1.63	0.15
Network Racial Composition	Exclusive or Predominant White	1	Ref						
	Racially Equal			1.12	0.46, 2.68	0.81	1.32	0.48, 3.64	0.60
	Exclusive or Predominant Black			1.43	0.49, 4.21	0.52	3.24	1.08, 9.73	*
Network Gender Composition	Predom. or Exclusively Male	1	Ref						
	Equal Male and Female			0.51	0.19, 1.34	0.17	1.29	0.49, 3.42	0.61
	Predom. or Exclusively Female			0.68	0.23, 1.95	0.47	0.55	0.15, 2.01	0.36

^a^ Reference category, * *p* < 0.05, ** *p* < 0.001

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
