# Peer review of "Constructing Taxonomies: Identifying Distinctive Class of HIV Support and Risk Networks among People Who Use Drugs (PWID) and Their Network Members in the HPTN 037 Randomized Controlled Trial"

_ijerph, 2022, doi:10.3390/ijerph19127205_

Round 1

Reviewer 1 Report

The authors addressed my suggestions.

Reviewer 2 Report

Thank you, the authors have addressed all my concerns with sufficient detail. 

This manuscript is a resubmission of an earlier submission. The following is a list of the peer review reports and author responses from that submission.

Round 1

Reviewer 1 Report

In this substudy from a randomized clinical trial, Van Sluytman et al. identified risk clusters associated with injection drug use (IDU) among 232 HIV-negative individuals and the characteristics of their social networks. The authors identified 3 clusters, characterized by low risk (LR), high risk associated with the use of Paraphernalia (PR), and high risk associated with the use of Paraphernalia and high-risk sexual behavior (PRS). In addition, several individual and network risk factors for belonging to one of those groups were identified. 

The manuscript is clearly structured and very well written. The results of the study are important and help to advance our knowledge on risk behavior among people who inject drugs (PWID), thereby providing insights for prevention efforts. However, I have several concerns, of which the most important are the following: 

  1. Notion of risk: In paragraph “2.1 Measures”, it should be made more explicit what type of risk the questions are designed to address. I suppose the authors aimed to assess the risk for acquiring HIV. However, other infections including hepatitis C could be transmitted with IDU too, for which sexual transmission among heterosexual individuals is a less important driver. Also, judging from figure 1, the PRS risk group is characterized by unprotected sex with a primary partner. However, it is unclear whether this denotes a stable or unstable partner, a distinction that substantially influences the infection risk. This should be made more clear in the methods section, and probably discussed as well. 
  2. Information on network members: Please provide information on how information about network members was gathered. Is this reported by network members, or by the index individuals? If it was assessed by asking the index individuals, I wonder whether there is missing information (e.g. incarceration, highest education). Please clarify.
  3. Timepoints: In paragraph “2.2 Analyses”, the authors mention that data from baseline and at six months were used for the analyses. Which time point was used to derive clusters and perform the logistic regression models? Did you use both in case of missing data? It would be interesting to know whether these network allocations were stable over time (baseline = six months), but I assume the allocation was subject to change in response to the intervention you performed in-between. 

Minor comments:

  1. Title: Is very long and difficult to understand, and could benefit from removing some of the jargon words.
  2. Abstract: Please write out all abbreviations on first occurrence (esp. PR, PSR, LR etc.)
  3. Results: Please provide the number of individuals and networks from the USA and Thailand, respectively. 
  4. Line 198-201: It is unclear what the numbers in that paragraph signify. I assumed that AIC, BIC, and Entropy should be the same as in Table 3 in the column for “3 classes”?
  5. Figure 1: This figure is essentially unreadable because of the overlap of all the lines. Please label the y-axis and provide a legend with an explanation of what you show in the figure. To avoid the overlap of the lines at the “0” line, maybe you could use 3 facets side-by-side. Also, consider using a log10-scale for the y-axis to show the small differences more clearly. 
  6. Line 209-2011: “They responded affirmatively to lower rates of using discarded needles…” is very difficult to read, please reword (e.g. they were less likely to use discarded needles) 
  7. Line 215: “... with primary PARTNERS were higher than …”
  8. I might misinterpret some of the findings, but in lines 237-240 the authors mention that the networks comprising only black individuals increased the odds of membership in the low-risk class. However, in lines 266-269 they write that networks with predominantly black participants faced high odds of belonging to the paraphernalia and sex risk class. Please clarify. 
  9. Line 274-275: Incomplete sentence. 
  10. Lines 415-416 should belong to the limitations section. 

Author Response

  • Notion of risk: In paragraph “2.1 Measures”, it should be made more explicit what type of risk the questions are designed to address. I suppose the authors aimed to assess the risk for acquiring HIV. However, other infections including hepatitis C could be transmitted with IDU too, for which sexual transmission among heterosexual individuals is a less important driver. Also, judging from figure 1, the PRS risk group is characterized by unprotected sex with a primary partner. However, it is unclear whether this denotes a stable or unstable partner, a distinction that substantially influences the infection risk. This should be made more clear in the methods section, and probably discussed as well. 
    1. Revised per the reviewer comment: This project conceptualized risk as engaging in behavior associated with HIV transmission e.g., sharing rinse water or other drug use paraphernalia.
  • Information on network members: Please provide information on how information about network members was gathered. Is this reported by network members, or by the index individuals? If it was assessed by asking the index individuals, I wonder whether there is missing information (e.g. incarceration, highest education). Please clarify.
    1. Revised per the reviewer comment: Outreach workers recruited index participants. Index” participants met the following eligibility criteria age 18 or older, injected drugs at least 12 times in the prior three (3) months, not enrolled in methadone maintenance treatment in the past three (3) months, HIV negative antibody test results within 60 days before randomization, willingness to identify and attempt to recruit at least two eligible HIV risk network members and recruit at least one eligible risk network member. Network member eligibility criteria include age18 or older, recruited for the study by an eligible index participant, and injected drugs with or had sex with the index participant who recruited them within the previous three months. It employs data collected at baseline, and a 6-month follow up from both index and network members at baseline and follow-up interviews.
  • Timepoints: In paragraph “2.2 Analyses”, the authors mention that data from baseline and at six months were used for the analyses. Which time point was used to derive clusters and perform the logistic regression models? Did you use both in case of missing data? It would be interesting to know whether these network allocations were stable over time (baseline = six months), but I assume the allocation was subject to change in response to the intervention you performed in-between. 
    1. With all due respect, the authors are unclear on what is meant by “clusters” Networks were established at baseline. The class was derived from LCA, examining risk behavior at both points.

Minor comments:

  • Title: Is very long and difficult to understand, and could benefit from removing some of the jargon words.
    1. Unfortunately these are funder requirements.
  • Abstract: Please write out all abbreviations on first occurrence (esp. PR, PSR, LR etc.)
    1. Revised per the reviewer's comment.
  • Results: Please provide the number of individuals and networks from the USA and Thailand, respectively. 
    1. Revised per the reviewer's comment. Line 127
  • Line 198-201: It is unclear what the numbers in that paragraph signify. I assumed that AIC, BIC, and Entropy should be the same as in Table 3 in the column for “3 classes”?
    1. Revised per the reviewer's comment. LCA revealed a three-typology solution among participants. Model fit was determined by Akaike information Criterion 6130.05, Bayes information criterion 6288.42, and entropy 0.804. Presented below are the model fit criteria outcomes for three, four, and five classes.
  • Figure 1: This figure is essentially unreadable because of the overlap of all the lines. Please label the y-axis and provide a legend with an explanation of what you show in the figure. To avoid the overlap of the lines at the “0” line, maybe you could use 3 facets side-by-side. Also, consider using a log10-scale for the y-axis to show the small differences more clearly. 
    1. Per the reviewer's comment, I used 3 facets side-by-side, Negative or zero values cannot be plotted correctly on log charts. Only positive values can be interpreted on a logarithmic scale.
  • Line 209-2011: “They responded affirmatively to lower rates of using discarded needles…” is very difficult to read, please reword (e.g. they were less likely to use discarded needles) 
    1. Revised per the reviewer comment: They reported lower rates of discarded needles use and lower rates of engaging in high-risk sexual activity with primary and other partners than other groups.
  • Line 215: “... with primary PARTNERS were higher than …”
    1. Revised per the reviewer's comment" “Additionally, their rates of engaging in high-risk sexual activity with primary were higher than member of the Low Risk class.”
  • I might misinterpret some of the findings, but in lines 237-240 the authors mention that the networks comprising only black individuals increased the odds of membership in the low-risk class. However, in lines 266-269 they write that networks with predominantly black participants faced high odds of belonging to the paraphernalia and sex risk class. Please clarify. 
    1. Revised per the reviewer comment: Network demographic characteristics, i.e., belonging to a network comprising exclusively or predominantly Black members (OR = 1.35; 95% CI 0.72, 2.52) increased the odds of membership in the High Sex/Moderate Paraphernalia Risk Class
  • Line 274-275: Incomplete sentence. 
    1. Revised per the reviewer comment: For example, individuals with some college faced increased odds of belonging to Paraphernalia Risk Class (OR = 1.05; 95% CI .36, 3.08). In contrast, among members of the Paraphernalia and Sex Risk Class, participants with some college reduced their odds of belonging to the Paraphernalia and Sex Risk Class (OR =.77; 95% CI .23, 2.59).
  • Lines 415-416 should belong to the limitations section. 
    1. Please note the sentence is the final sentence of the paragraph beginning. "Our data faces several limitations. As a secondary data analysis

Reviewer 2 Report

Introduction: The article presents various literature gaps and does bring something new to the literature. The introduction contains relevant information and adequately speaks about prior studies and what is already known about the topic to point out the gap in the literature. The lead in sentence of each paragraph does a great job at summarizing the contents of each paragraph, allowing the reader to follow along and remain engaged. The introduction clearly states the objectives and the aims are concisely spelled out in the final paragraph. The separation of the section about taxonomies specifically was well placed and gives the audience a strong foundation to understand the sociology that will be explored in this population. A clear scope of this study is defined.

Method: This study is a secondary data analysis of HPTN 037, a phase III, multi-site, two-arm, randomized controlled trial. That study included roughly 2,610 HIV uninfected injection drug users (index participants) and individuals they identified as being part of their sex and drug-using networks (network members). These participants were randomized into one of two groups in a 1:1 ratio. Both arms received basic HIV counseling and testing enhanced with interactive risk reduction counseling. Index participants randomized to the experimental arm also received peer educator training. The purpose of HPTN 037 was to determine the efficacy of a network-oriented peer educator intervention for prevention of HIV infection among PWID and members of their HIV risk network though reduction of HIV risk behaviors.

This secondary analysis draws its sample from the 232 networks enrolled in Philadelphia, which included 232 index participants and 464 network members. The data studied was collected at baseline and a 6-month follow up. Injection risk was assessed with a series of questions regarding sharing materials used to inject drugs, the location in which drugs were injected, and sexual behaviors. Participants answered using a dichotomous variable. Statistical software was used to analyze the data. A univariate analysis measured the variables within the sample. Latent class analysis was then used to find su

Some concerns of this analysis:

  • Less than half the participants of HPTN 037 were included
  • Results could be biased for Philadelphia
  • The dichotomous variables were opposite for the sets of questions: 0 indicated “never/no” and 1 indicated “one or more times/yes” for the questions regarding injecting drugs, but 0 indicated “yes” and 1 indicated “no” for the questions regarding sexual behaviors.
  • Only followed participants for 6 months
  • Was it necessary to use both Bayesian information criterion and Akaike’s information criterion?
  • Was multinomial logistic regression appropriate for dichotomous variables?

Results: Three risk latent classes were derived: Low Risk (LR), Paraphernalia Risk (PR), and Paraphernalia/Sex Risk (PSR). When describing the individual and demographic characteristics of each group, the wording was confusing. The flow is interrupted as the author constantly flips back and forth between what increases and decreases the odds of membership in one of the classes, making it hard to follow.

Discussion: The author succeeded in providing a coherent discussion about the findings and the associated endpoints. The discussion delivered well on the implication of the findings along with the limitations within the study design and analysis.

Conclusion: Even though the conclusion is adequate in length, the author failed to make the content specific. The content is very broad and does not truly conclude the findings and purpose of the study. The conclusion also tells readers multiple facts that are common sense or widely known by the audience.

Author Response

Review 2

Some concerns of this analysis:

  • Less than half the participants of HPTN 037 were included
    1. to ensure the ability to a contextual potential finding the sample was limited to the US networks that were recruited in Philadelphia
  • Results could be biased for Philadelphia
    1. With all due respect, Philadelphia is a very large and diverse city, representative of other urban communities in the United States of America.
  • The dichotomous variables were opposite for the sets of questions: 0 indicated “never/no” and 1 indicated “one or more times/yes” for the questions regarding injecting drugs, but 0 indicated “yes” and 1 indicated “no” for the questions regarding sexual behaviors.
    1. This variation assisted in the determination of the referent group
  • Only followed participants for 6 months
    1. This is not true. However, we only used the data from the base line and six-month interviews
  • Was it necessary to use both Bayesian information criterion and Akaike’s information criterion?
    1. Bayesian information criterion and Akaike’s information criterion are critical steps in validating the LCA.
  • Was multinomial logistic regression appropriate for dichotomous variables?
    1. the outcome variable for the multinomial regression was the likelihood of belonging to three risk classes.
  • Results:Three risk latent classes were derived: Low Risk (LR), Paraphernalia Risk (PR), and Paraphernalia/Sex Risk (PSR). When describing the individual and demographic characteristics of each group, the wording was confusing. The flow is interrupted as the author constantly flips back and forth between what increases and decreases the odds of membership in one of the classes, making it hard to follow.
    1. In following the outcomes for the groups, it was essential to articulate variation in risk with all due respect. This results in comparing and contrasting variations that, at times, were greater and lesser. To cluster the findings into greater and lesser risk one compromise the readers' ability to easily access the variations
  • Discussion: The author succeeded in providing a coherent discussion about the findings and the associated endpoints. The discussion delivered well on the implication of the findings along with the limitations within the study design and analysis.
  • Conclusion: Even though the conclusion is adequate in length, the author failed to make the content specific. The content is very broad and does not truly conclude the findings and purpose of the study. The conclusion also tells readers multiple facts that are common sense or widely known by the audience.
    1. With all due respect, the authors sought to 1) identify taxonomies of networks among HIV negative people who use drugs (PWID) and members of their HIV support and risk networks, 2) determine the characteristics of the network, and 3) determine the impact of the individual and network-level factors on class membership. These items were each identified and associated with larger concepts that though widely known, are not often considered from this lens.
